# ICIF: Image fusion via information clustering and image features

Linlu Dong[1], Jun Wang[1]*, Liangjun Zhao[2,3], Yun Zhang[2], Jie Yang[1]

**1** School of Information Engineering, Southwest University of Science and Technology, Mianyang, Sichuan, China, **2** School of Computer Science and Engineering, Sichuan University of Science and Engineering, Zigong, Sichuan, China, **3** Sichuan Key Provincial Research Base of Intelligent Tourism, Sichuan, China

* 1007380441@qq.com

**Data Availability Statement:** All relevant data are available from the article.

**Funding:** Sichuan Key Provincial Research Base Project of Intelligent Tourism(NO:ZHZJ22-03). Sichuan Science and Technology Program:

## Abstract

Image fusion technology is employed to integrate images collected by utilizing different types of sensors into the same image to generate high-definition images and extract more comprehensive information. However, all available techniques derive the features of the images by utilizing each sensor separately, resulting in poorly correlated image features when different types of sensors are utilized during the fusion process. The fusion strategy to make up for the differences between features alone is an important reason for the poor clarity of fusion results. Therefore, this paper proposes a fusion method via information clustering and image features (ICIF). First, the weighted median filter algorithm is adopted in the spatial domain to realize the clustering of images, which uses the texture features of an infrared image as the weight to influence the clustering results of the visible light image. Then, the image is decomposed into the base layer, bright detail layer, and dark detail layer, which improves the correlations between the layers after conducting the decomposition of a source graph. Finally, the characteristics of the images collected by utilizing sensors and feature information between the image layers are used as the weight reference of the fusion strategy. Hence, the fusion images are reconstructed according to the principle of extended texture details. Experiments on public datasets demonstrate the superiority of the proposed strategy over state-of-the-art methods. The proposed ICIF highlighted targets and abundant details as well. Moreover, we also generalize the proposed ICIF to fuse images with different sensors, e.g., medical images and multi-focus images.

## Introduction

The fusion of various types of images effectively supplements the defect and limitation induced in a single sensor. Due to this advantage, a promising system is considered and has been widely studied in many applications including video images [1–5], industrial fields [6], medical fields [7, 8], security monitoring [9], digital photography [10] and remote sensing fields [11, 12].

Due to the demands from many application areas, image fusion technology has been actively studied and rapidly advanced. The image fusion methods are categorized into two groups: spatial domain fusion and transformation domain fusion [13]. The spatial domain fusion algorithms directly used the pixel value of the image as the fusion rule. These methods

(NO:2023YFS0371). The funders had no role in study design, data collection and analysis, decision to publish, or preparation of the manuscript. Liangjun ZHAO are the recipient of the funding awards listed above.

**Competing interests:** The authors have declared that no competing interests exist.

have low computational complexity and are thus efficient in processing a large number of images. However, it was more challenging to segment the salient feature areas into infrared images, which led to the lack of anti-jamming capability and mass information loss. On the other hand, the fusion methods in the transformation domain applied a multi-scale decomposition to scale the multi-layer spatial feature information of the image overlay [14] and obtain different scaled spaces. In this way, the infrared image and the visible image were effectively fused. The available image fusion methods can be divided into conventional methods and deep learning-based methods.

The conventional methods include (1) multi-scale algorithms, such as Wavelet [15], Biorthogonal wavelet [16], MDLatLRR [17], Guided filtering [18], Nonsubsampled shearlet transformation (NSST) [19]. The multi-scale algorithm can decompose the detailed entire image by transforming it from the spatial domain to the frequency domain and then obtaining the result through a fusion strategy. However, the simultaneous interpretation of images using different sensors often requires different multi-scale decomposition algorithms. Also, during the transformation process, it may cause an unrecoverable loss of data. (2) Dictionary learning methods include JSRSD [20], DDL [21], Sparse K-SVD [22], DLLRR [23], TS-SR [24], DCST-SR [25], and ConvSR [26]. Unlike multi-scale methods, they do not suffer from the loss of image details since the transformation to the frequency domain is not required. However, dictionary learning has usually higher computational complexity. Further, the fusion performance is limited to the complex texture of images.

Besides, the title called deep-learning methods include both Machine learning (ML) and Deep learning (DL) models that have been broadly implemented in many areas such as image fusion [27, 28], agricultural surveillance [29, 30], environmental monitoring [31–36], sentiment analyses [37], medical image processing [38], and cyber security [39–41]. Moreover, the implementation of the deep learning methods can be divided into (1) training models, and (2) non-training models. (1) The algorithms that need to train the network model include CNN [42], Unsupervised [43], DneseFuse [44], FusionGAN [45], and IFCNN [46] when image fusion is under consideration. With a well-designed network structure, no hand-crafted strategies are required to achieve good fusion results. However, a vast amount of data specified to the target fusion tasks are required, which often does not perform well when different types of data are available. For example, the fusion model trained with bright images does not work well on low-illumination images. (2) The non-training models include VggML [47] and ResNet-ZCA [48]. In those methods, pre-trained deep neural networks extracted deep multi-level features used for image fusion tasks. The deep multi-level features contain richer information which is beneficial for image fusion operations. Since those pre-trained networks are not fine-tuned to the target task, their robustness is limited.

Specifically, fusion methods involve three crucial challenges, i.e., image transformation, activity-level measurement, and fusion rule designing [49]. The three constraints have become increasingly complex, especially for designing fusion rules in a manual way which strongly limits the development of the fusion methods. To address the abovementioned issues, researchers use different theories to obtain results with a richer texture and significant targets through the analysis of advanced fusion algorithms. Considering the characteristics of human vision, the multi-scale theory has become a hot research topic for image fusion. Although the fusion algorithm based on multi-scale theory started as one of the initial methods, it has been gradually overtaken by the theory based on deep learning. In the final analysis, the main reasons why there is no breakthrough in the available multiscale theory are as follows: 1. the infrared and visible images are decomposed into detailed and basic parts in the frequency domain, resulting in partial information loss and increased computational complexity. 2. the implemented fusion rules such as the average fusion rule [50], gradient difference fusion rule [51],

and so on, will cause the loss of useful information with weak intensity [52], and the generalization ability of the fusion effect is also poor. 3. to extract image features, the information is only derived under the influence of the internal features of the respective image, without considering the influence of external features on the internal features, resulting in the poor correlation of various feature layers extracted from the two source maps to be fused. In the fusion process, the selection of a fusion strategy is difficult to take into account all the differences, resulting in the loss of some information, which further results in the poor clarity of the generated fusion images. To this end, the ICIF algorithm is proposed.

More precisely, the fusion algorithm can directly process the pixels of the image to be fused in the spatial domain so that the original information can contribute to the process of image fusion, reduce information loss in the transformations between the frequency domain and spatial domain, and reduce the computational complexity. Therefore, the necessity of using image clustering to achieve the task of multi-scale decomposition of an image is tenable. However, the image clustering methods selected by employing similar multi-scale fusion technologies have mainly focused on the smoothness of images. Besides, it ignores the layers obtained from multi-scale decomposition in the fusion task and requires reconstruction to generate a new fusion image. Therefore, when an image clustering is run, the features of the two sources of images affect each other, and the extracted fusion image can have more relevance. At the same time, the weight of the fusion strategy should also consider the characteristics of the images collected by using different sensors and the characteristics of each layer to determine the fusion weight. So, a new image with higher clarity and more comprehensive information can be generated.

On the other hand, it is impossible to improve the robustness of the fusion algorithm by relying only on the fixed weight or the features of the layer itself as the weight. Fig 1 shows an illustration to present the core idea of the proposed method that is compared with the state-of-the-art multi-scale and neural network-based fusion methods. The result of the proposed ICIF has both high-definition and intuitive information expression. While the MDLatLRR is a whole fuzzy approach, the FusionGAN can maintain a large amount of infrared information but lose the visible texture information.

Given specific texture details enable us to quickly judge the vehicle that is a camouflaged car according to the body texture with the proposed ICIF. However, it is difficult to judge the vehicle that is a camouflaged car with the fusion results of the MDLatLRR and the FusionGAN. Also, the MDLatLRR and FusinGAN provide fuzzy ground texture, where the structure of the road surface is difficult to be judged.

Our contributions include the following three aspects. (*i*) we propose a new method to realize the strong correlation between the proposed feature layers through the mutual influence of the feature information of two sources of images. Besides, it makes the extracted feature layers include both the features of infrared images and visible images concurrently and improves the

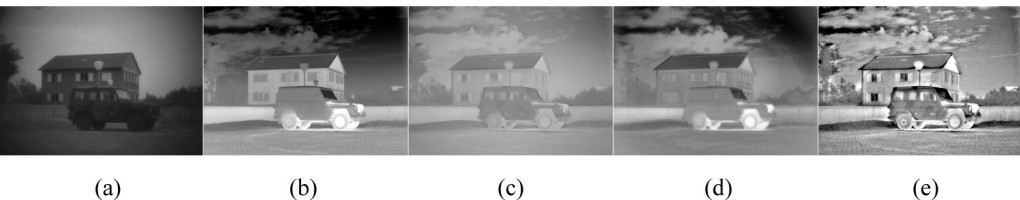

| (a) | (b) | (c) | (d) | (e) |

**Fig 1. The comparison of the fusion results.** (a)Visible image (VIS), (b) Infrared image (IR), (c) MDLatLRR [17], (d) FusionGAN [45], (e) ICIF.

consistency between each layer. As far as we know, the utilization of this correlation-based clustering method to achieve multi-scale decomposition of images has not been studied yet. (*ii*) we have explored the characteristics of images collected by using different sensors through experiments and found that the image characteristics collected by utilizing different sensors meet certain rules. According to this rule, when combined with the layer characteristics of multi-scale decomposition, we have formulated a novel fusion strategy, which makes the proposed fusion algorithm robust to fusion tasks in different fields. (*iii*) we provide qualitative and quantitative comparisons utilizing several advanced methods in publicly accessible datasets. Furthermore, when compared with previous methods, the proposed method can produce fusion results of sharpened infrared images with prominent targets and rich textures.

The rest of the paper is organized as follows. Section 2 illustrates the relevant work of multi-scale algorithms, highlighting the novelty of the proposed fusion method. Section 3 introduces the information clustering algorithm, the mathematical model of the acceleration algorithm, and the mathematical model of the guided image and introduced the multiscale decomposition process of images. In Section 4, the novel fusion strategy of images is introduced. Through experimental conduct, the characteristic rules of different sensors are explored, and the final fusion weights are combined with the decomposed layer features to generate the final fusion results. Section 5 is allocated to the outcomes of the experiments and discussion. Section 6 finally concludes this paper.

## Related work

At present, the fusion technology based on a multi-scale framework transformation of the image from the spatial domain to the frequency domain proposes the features of an image in the frequency domain, reconstructs these features through the fusion strategy, and finally transforms the image from the frequency domain to the spatial domain to complete the image fusion process. Ma [42] and others claim that different sensors have different imaging principles, and it is one-sided to use the same frequency-domain transformation technology to extract image features. Besides, they proposed a new fusion algorithm based on gradient transmission and the minimization of the total change. Although the fusion results of this method maintain the significance of infrared rays, the fusion results lost a lot of texture of visible images [35]. Bavirisetti et al. [53] proposed a method based on anisotropic diffusion and the Karhunen-Loeve change. The source image is smoothed by various anisotropic diffusion methods. After the image is decomposed into base and detail layers, the image is reconstructed by the Karhunen-Loeve orthogonal change. Although this method can maintain the texture of the visible image and the target information of the infrared image due to the insufficient decomposition layers and the fixed fusion strategy, the generated fusion results are fuzzy and poor in clarity. Zhao et al. [5] proposed a method to decompose the image into the base layer, the bright detail layer, and the dark detail layer, and then use the features of the layers as the fusion weights. This technology can not only fully extract the feature information of each source image, but also realize the adaptive ability of the algorithm according to the features of the layers. The clarity of the generated fusion image is greatly improved consequently. However, the above-mentioned technologies for image fusion in the airspace do not consider the mutual relationship between the images to be fused. Such a relationship is just like the accessories on the car. If components from different brands of cars are used to assemble a new car, they highly likely could not be assembled successfully because of the different sizes and specifications of the components. Even if a new car is forcibly assembled, its stability and safety would be greatly reduced. Therefore, the fundamental principle of the image fusion process is the same as that of an automobile assembly. In the process of extracting layers, the texture

features of each source image should be fully considered to improve the consistency of each layer, so that the fusion process will not cause information loss due to layer differences. At the same time, exploring the characteristics of images collected by using different types of sensors, and combining the characteristics of each layer leads to developing a fusion strategy that can further improve the convergence of fusion and improve the robustness of the algorithm in the manuscript.

## Formulation of the multi-scale decomposition method

The key to the success of the multi-scale image fusion method is based on extracting image scales. Typically the implemented scale extraction algorithms include anisotropic diffusion [53] and low-rank representation (LRR) [54]. However, these methods have great differences in the extracted outcomes of different homologous images when extracting the detailed layer of images and not retaining the significant features in the image.

### Motivation

To clearly express the novel points of the proposed approach, we use a viewable approach to show the relevant work (ADF [53]) and how the proposed method differs is presented. Among them, the clustering of the ADF is shown in Fig 2(a), and the proposed clustering process is shown in Fig 2(b).

To further dissect the difference between the proposed technique and the available techniques, the surf plots of some regions of the clustering results based on the 'AD base image' and 'IC base image' in Fig 2 are plotted separately as shown in Fig 3.

Fig 3 depicts a surf plot of the local region of the infrared image, with most of the texture details highlighted on the surface. In the surf diagram of the local region of the *AD* base image, although the surface is smooth, there is also a large amount of bulge information between the peak and the trough, such as at the green circular box. The corresponding *AD* base image also retains a large amount of significant target information, such as the human leg information at the red arrow, which indicates that this part of the information in the decomposed detail feature layer has not been extracted. When the clustering process is considered, the proposed method adopts the texture feature of the visible light image as the guide map, so that the prominent texture at the green circular box in the surf plot in Fig 3(c) is smoothed because the low texture gradient at the corresponding position of the visible light image in Fig 3 expands the degree of smoothing, which improves the correlation between the infrared and the visible light images regarding feature extraction process. Therefore, the detailed feature layer obtained after the decomposition contains this partial information.

### Information clustering algorithm

The information clustering algorithm computes the current pixel value with the median of the weighted neighboring pixel values in the local window. The current pixel is defined to be $p$ in the $I$ image and the convolution window is defined to be $R(p)$ of radius $r$. For all pixels $q \in R(p)$, where the weight $W_{pq}$ is determined according to the affinity of the pixels $q$ and $p$ in the guidance map $f$.

$$W_{pq} = g(f(p), f(q)), \qquad (1)$$

where $f(p)$ and $f(q)$ are the intensity values of pixels $p$ and $q$ in the guidance map, respectively, $f$. $g$ represents the influence function between two pixels, defined by

$$g = e^{-||f(p)-f(q)||}, \qquad (2)$$

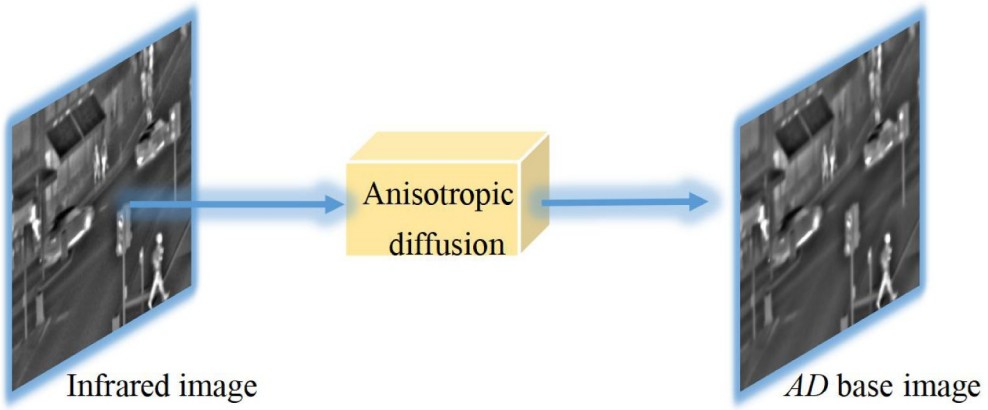

(a) Clustering results and technical process of the ADF algorithm

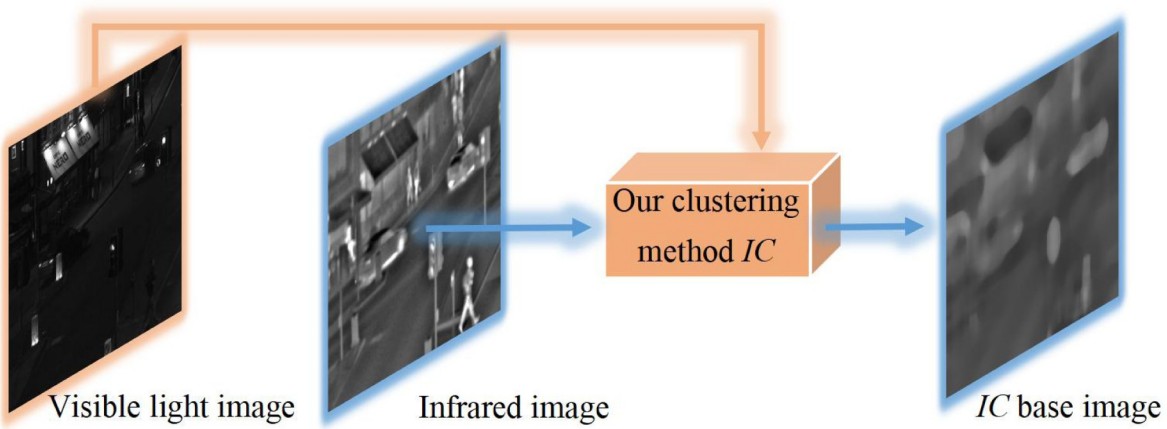

(b) Clustering results and technical flow of the proposed algorithm

**Fig 2. Comparison of the existing art and the proposed technology.** (a) Clustering results and technical process of the ADF algorithm. (b) Clustering results and technical flow of the proposed algorithm.

Let $I(q)$ denote the value of $q$ in image $I$. $n = (2r + 1)^2$ represents the number of pixels in $R(p)$. Then, the values are sorted in ascending order, and the central pixel $p$ of the convolution window is weighted to obtain a new pixel value $p^*$. That is, $I(p)$ is modified to $I(p^*)$ in the image. The mathematical model of $p^*$ is given as follows [55]:

$$p* = \min k \ \text{ s.t. } \ \sum_{q=1}^{k} w_{pq} \geq \frac{1}{2} \sum_{q=1}^{n} w_{pq}, \tag{3}$$

For all pixels before the computation of $p^*$, this definition means the sum of corresponding weights should be approximately half of all weights summed together. $f$ map in Eq (6) determines weights. Practically, $f$(p) can be intensity, color, or even high-level features depending on the problem definition.

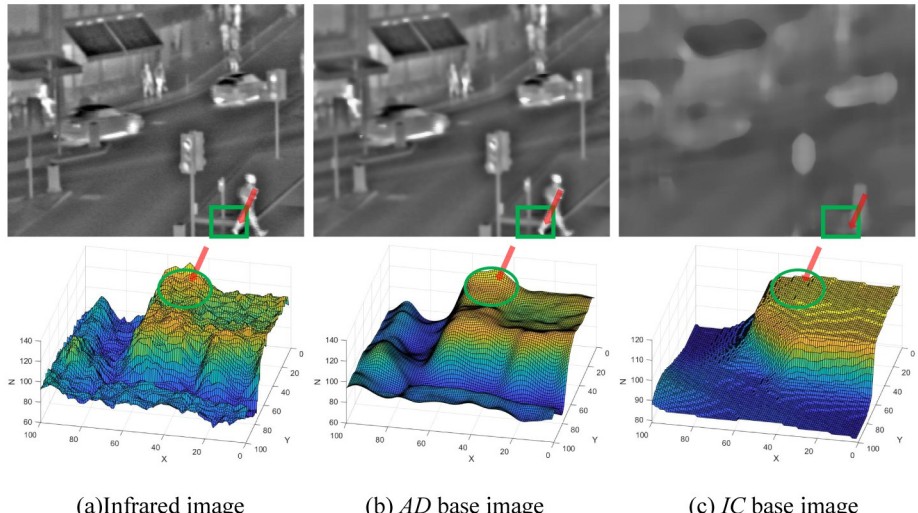

(a)Infrared image (b) *AD* base image (c) *IC* base image

**Fig 3. Surf map of some regions of image clustering results.** (a) The upper row represents the infrared image, and the lower row represents the surf map of the corresponding area of the green rectangular box. (b) The upper row shows the clustering effect of the anisotropic diffusion (*AD*) method, and the lower row shows the surf map of the corresponding area of the green rectangular box. (c) The upper row shows the clustering effect of our clustering (IC) method, and the lower row shows the surf map of the corresponding area of the green rectangular box. While the X-axis represents the row coordinates of the pixel points in the image (set value range: 0 ~ 100), the Y-axis represents the column coordinates of the pixel points in the image (set value range: 0 ~ 100), and the Z-axis represents the values of the pixel points corresponding to each coordinate point (value range is determined by the source image).

The weight W is recomputed whenever the weighted median filter updates the central pixel of the convolution window. Thus, the computational complexity of the weighted median filter would be too high to be applied in practical engineering applications. Therefore, an accelerator is designed in this paper. Each pixel in the image is represented by an original two-dimensional coordinate $(I(q), f(q))$, where $I(q)$ and $f(q)$ represent the intensity value of the image and the characteristics of the image, respectively. For an $n*n$-dimensional convolution window, $Ni$ intensity values and $Nf$ image features within the window are assumed. The image intensity value $i$ is used as the abscissa parameter, and the guidance feature $f$ is used as the ordinate parameter to form a joint histogram $H(i, f)$. It is formulated as follows [55]:

$$H(i,f) = \#\{q \in R(p)|I(q) = I_i, f(q) = f_f\}, \tag{4}$$

where # denotes the number of eligible elements.

For all image pixels corresponding to coordinates $(i, f)$, the weight is computed as $g(f_f, f(p))$ according to the obtained joint histogram $H(i, f)$ by considering the center point pixel $p$ in the convolution window. Finding the weight of pixel $p$ is reformulated by finding the sum of the weights of $H(i, f)$ pixel $i$ by traversing the joint histogram, as follows:

$$W_i = \sum_{f=0}^{N_f-1} H(i,f)\, g(f_f, f(p)), \tag{5}$$

where $g(f_f, f(p))$ represents the pixel weight with the feature $f$ and the central pixel $f(p)$.

Computing a median value consists of two sub-calculations. First, the weights are accumulated from left to right to obtain the weight and $W_t$. Then, the obtained weights are accumulated again from left to right. When the sum of weights is obtained, which is approximately

matched to the half of the $W_t$ value of the first calculation, the corresponding $i$ value is the median value of the convolution window. The value of the current pixel is replaced by the $i$ value, completing the median filtering process. However, the median search process has higher computational complexity. To reduce search time, Cut Points and Balance approaches were introduced. The weights of all pixels in the convolution window are arranged at pixel $i$, and the weights on the left and $W_l$ are compared with those on the right. The weight and $W_r$ should be equal. The Cut point is the pixel point $i$ where the formula holds $W_l - W_r = 0$, and represents balance. The Cut point is solved by minimizing the following objective function defined by

$$\min |W_l - W_r| \text{ s.t. } W_l - W_r \notin \mathrm{R}^-. \tag{6}$$

To further reduce the calculation complexities for Cut Point in the whole image, the balance between the convolution windows with $p$ and $p+1$ as the center pixel is used. If the balance is positive, the Cut Point at $p$ is used as the Cut Point at $p+1$.

The abscissa of the joint histogram is accumulated and summed, and the entire joint histogram becomes a column matrix with $N_f$ elements. This process is called a 'balance counting box (BCB)' and is formulated as follows:

$$\begin{aligned} B(f) \quad &= \#\{q \in R(p)|I(q) \leq c, f(q) = f_f\} \\ &\quad -\#\{r \in R(p)|I(r) > c, f(r) = f_f\} \end{aligned}, \tag{7}$$

The BCB for the entire convolution window is defined as follows:

$$b = \sum_{f=0}^{N_f-1} B(f)g(f_f, f(p)). \tag{8}$$

Found that when constructing the BCB of the joint histogram, there will be a large number of 0 elements, and this will be also an influencing factor affecting the computational speed of the algorithm. The Necklace Table [55] method is used to skip the access of 0 factors and achieve effective acceleration of traversal.

The guidance map $f$ is a key factor in extracting texture in the ICIF. Our goal is to extract as much information as possible from the images to be fused from different sources when extracting the detail layer. To this end, this paper proposes a fuzzy combination method, where the high-frequency parts of the images from different sources are combined and normalized to form a feature map $\boldsymbol{f}$. This process is formulated as follows:

$$\boldsymbol{f} = 0.5 \times (I^*), \tag{9}$$

where $I^*$ indicates the visible light image when clustering the infrared images. Conversely, when the visible light image is clustered, then $I^*$ is the infrared image. Therefore, we record the clustering information as $IC(I)$ representing the base layer extraction of a given image $I$.

## Detailed layer extraction and characterization

Found that the detail layer of the images from different sensors can be divided into bright and dark detail layers. Also, the impact of the fused image of the bright detail and dark detail layers under the influence of different weights is high. Thus, the extraction of layers with different types of details and the fusion strategies have become an essential part of the successful implementation of the proposed method ICIF.

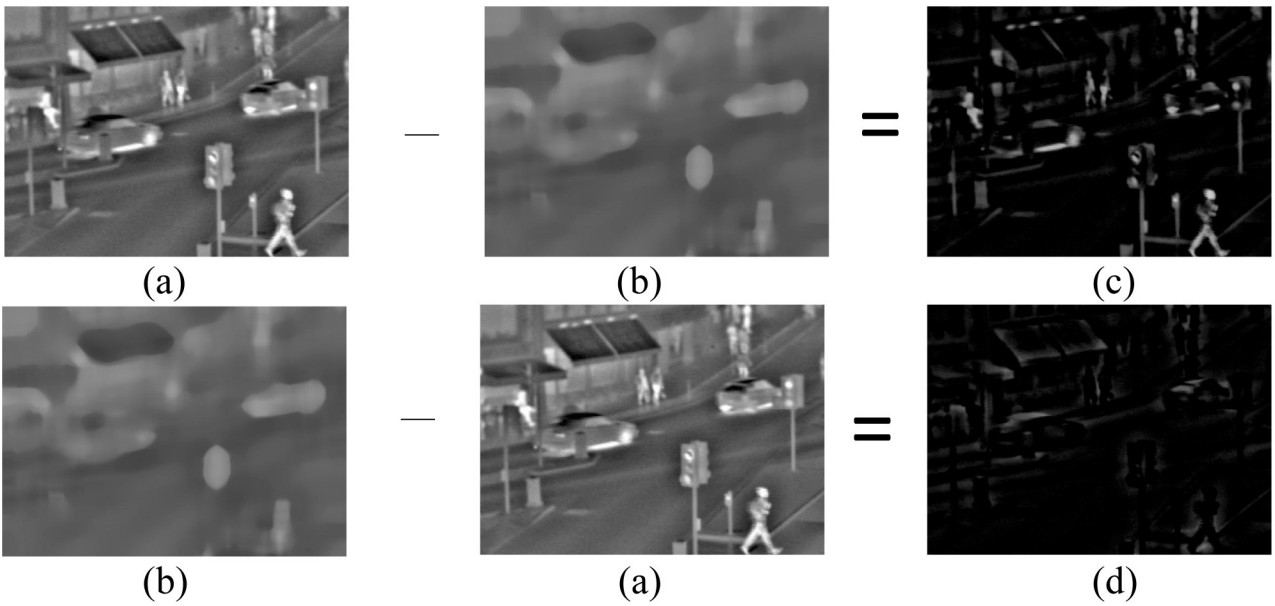

**Fig 4. The schematic diagrams of bright detail and dark detail extraction.** (a) Input image (b) Base-layer (c) Dark detail layer (d) Dark detail layer.

The bright detail layer, $I_{BD}$, is obtained by subtracting the base layer from the source image as follows:

$$I_{BD} = I - IC(I), \tag{10}$$

The dark detail layer, $I_{DD}$, is obtained by subtracting the source image from the base layer as follows:

$$I_{DD} = IC(I) - I, \tag{11}$$

Schematics of bright and dark detail extractions are shown in Fig 4.

## Fusion strategy

The sharpness of the fused image correlates with the size of the texture gradient of the image. Therefore, when the fusion strategy is developed, the realization of the fusion image can make full use of the infrared image and visible light image texture details, thus, a high definition is generated, and richer details in a fusion image are contained, according to the principle of expanding the gradient difference of a reconstructed fusion image. In other words, to base as the template, a brighter texture layer is superposition to the base, so the darker texture layer is portrayed to the base. However, to maintain the overall texture distribution of the generated images in harmony, we also need to formulate the weights in the processes of both superposition and characterization. Therefore, we studied two categories in the development stage of the fusion strategy. While the first is called the method of the base fusion, the second is called the weights of the detail layer and the base fusion.

## Basic layer integration strategy

To improve the texture structure correlation of the two images for the fusion of the base layer, we consider the method of taking the maximum $I_{Bmax}$ and the minimum $I_{Bmin}$ of image pixels

to fuse the two base layer images. Its mathematical model meets the following requirements defined in Eqs (12) and (13).

$$I_{B\max} = \max\{IC(I_1), IC(I_2)\}, \tag{12}$$

$$I_{B\min} = \min\{IC(I_1), IC(I_2)\}, \tag{13}$$

where $IC(I_1)$ and $IC(I_2)$ represent the base layer of the two input images, respectively.

To further strengthen the correlation of the base images, we consider taking the maximum image $I_{B\max}$ and the minimal image $I_{B\min}$ of the base images according to the standard deviation of each base image as the weights as follows:

$$I_{BF} = \frac{\min(\sigma_{B\max}, \sigma_{B\min})}{\sigma_{B\max} + \sigma_{B\min}} I_{B\max} + \left(\frac{\max(\sigma_{B\max}, \sigma_{B\min})}{\sigma_{B\max} + \sigma_{B\min}}\right) I_{B\min}, \tag{14}$$

where $I_{BF}$ represents the fused base layers of the two source images. $\sigma_{B\max}$ and represent the $\sigma_{B\min}$ standard deviations of $I_{B\max}$ and $I_{B\min}$, respectively. We use the multiple relationships of the standard deviations to indirectly reflect the relationship of the information intensity at the grassroots level. This addresses the different problems in results due to the different order of the sources of the input graph.

For quantitative analysis of the influences of different fusion strategies at the grassroots level on the final fusion result, the weight of the layer of fusion was set to 1, and the final fusion results obtained by employing different grassroots fusion strategies were analyzed. According to Eqs (12) and (13), the fuzzy maximum image and the fuzzy minimum image of the two image base layers, denoted by Imax, and Imin, were extracted.

Experiment 1: Only $I_{Bmin}$ as the base layer image.

Experiment 2: Only $I_{Bmax}$ as the base layer image.

Experiment 3: $I_{Bmin}$ and $I_{Bmax}$ as the base layer images after fusion according to the weight of Eq (14).

Fig 5 depicts the fused images according to different base fusion methods.

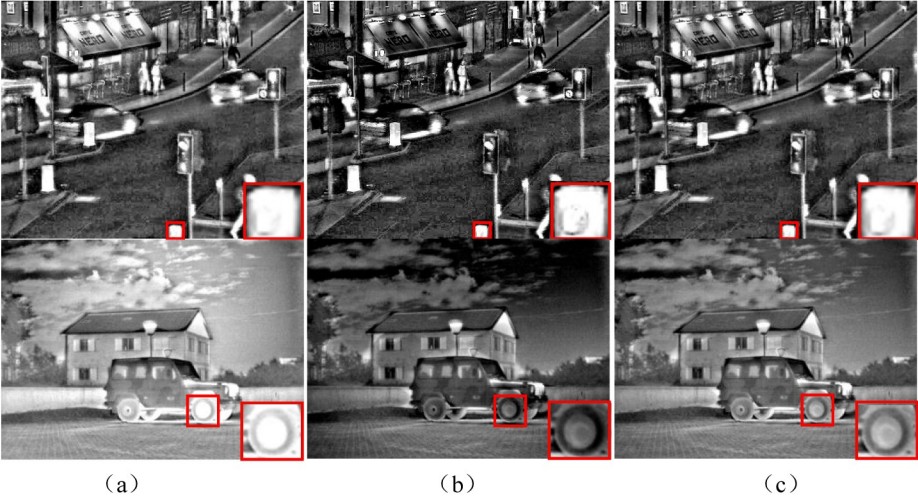

（a） （b） （c）

**Fig 5. The effect of different fusion strategies of grassroots images.** (a) Experiment 1, (b) Experiment 2, (c) Experiment 3.

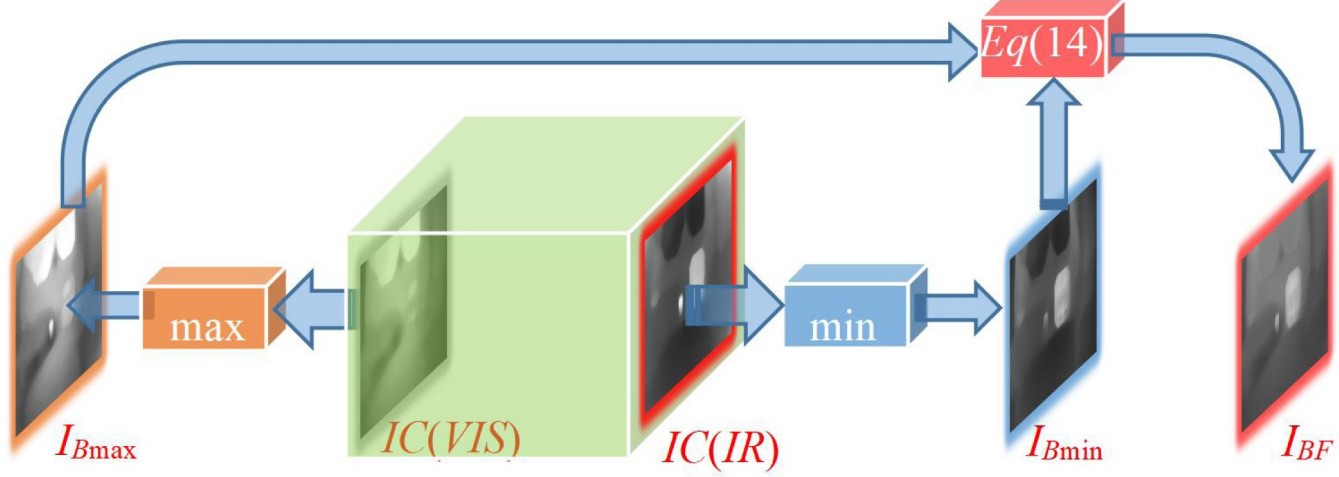

**Fig 6. Technical route of grassroots integration.**

Fig 5 compares the final fusion images obtained by the three base-layer strategies. Noted that the detail layer fusion weight was set all to 1. In Experiment 1 (Fig 5(a)), the brighter fused image is obtained, but part of the texture is covered by intense light. In Experiment 2, a strong ability to reproduce the texture of the image is achieved, but the whole image is relatively dim. The result of Experiment 3 is in between other strategies in texture reproduction and light intensity, and the scene in the image is more natural. After comparing the three experiments, Eq (14) is used as the fusion strategy of the ICIF basic image.

To demonstrate the process of the base fusion, we will provide the technical route, as shown in Fig 6.

## Determination of detail layer fusion strategy and weights

When the detail layer fusion weight is fixed to 1, no matter how the fusion strategy of the grass-roots changes. So, the fusion results cannot be comparable to the obtained images in Fig 2, confirming the necessity to study and discuss the fusion strategy at the detail level.

Unlike the base layer, the fusion of the detail layers is complicated because it needs to consider the images collected by using different sensors. Therefore, the algorithm adaptively assesses whether the fusion weight or the fusion method is determined according to the input image. According to the current multi-scale fusion algorithm strategy, the detail layer is mostly fused with the base layer according to a weight of 1, so it is difficult to obtain better fusion results in terms of the fusion effect.

The manuscript proposes an adaptive weight fusion scheme that adjusts the weights according to the sum of the square root of the information between the two source types of images and the maximum and minimum base information. In this way, the information from the two source types of images complements and cancels each other, resulting in the amount of information obtained being less than the sum of the two images. Therefore, to ensure algorithm efficiency, we use a squared extraction to fit the amount of information in the offset part. When the detail layer is merged with the base layer, the purpose of excessive enhancement will not appear.

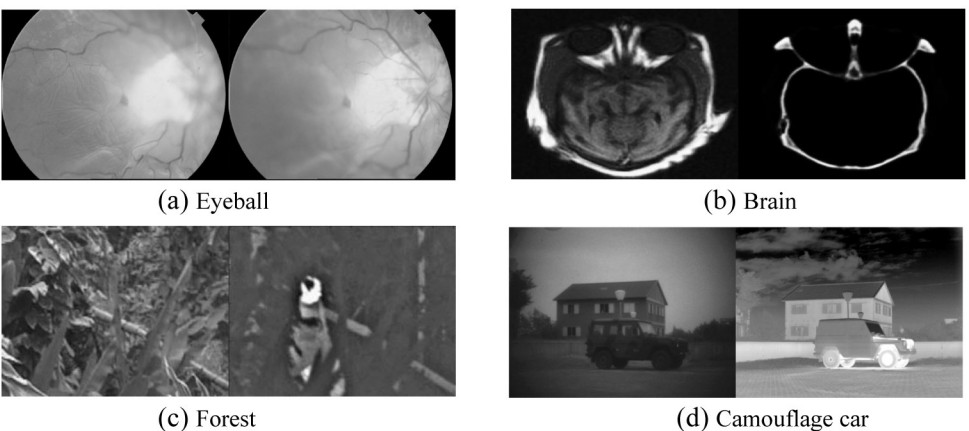

**Fig 7. The research objects of the image parameters.**

Specifically, the input image is decomposed into three layers, namely, a base layer, a bright detail layer, and a dark detail layer in the proposed method. Unlike the other multi-scale fusion algorithms, the fusion strategies for both the bright detail and dark detail layers need to consider the influence of the final fusion result. By aiming to expand the texture spacing, addition is used for fusing the bright detail layer with the base layer, while subtraction is used for fusing the dark detail layer with the base layer. The image fusion is formulated as follows:

$$I_F = I_{BF} + k(I_{BDC1} + I_{BDC2}) - k(I_{DDC1} + I_{DDC2}), \qquad (15)$$

where $I_F$ represents the fused image, both $I_{BD1}$ and $I_{BD2}$ are the bright detail layers of the two images, while both $I_{DD1}$ and $I_{DD2}$ are the dark detail layers of the two images, and $k$ represents the weight.

When the weights are all equal to 1, as shown in Fig 6, it can be judged that such a fusion rule is not optimal. To explore the relationship between image feature parameters and image fusion, we designed a group of experiments. The experimental objects are shown in Fig 7, and the corresponding feature parameters are shown in Table 1.

Table 1 summarizes that the characteristics of the source map and base layer can be obtained. In the fusion process, the feature of the detail layer can be obtained indirectly by subtracting the base feature from the feature of the source image. Therefore, the feature ratio of the detail layer to the base layer reflects the fusion process of the detail and base layers. Noted that after the source image is merged with the source image, its feature numbers are less than the sum of the two images. Similarly, the basic features are also the same. To fit this process, we use the form of a root sign. Therefore, we guess whether the algorithm can be adaptive by associating the fusion weight with the image eigenvalue. Thus, we have designed a weighted expression that is positively and negatively correlated with weight 1, and its expression is

**Table 1. Pixel attributes of each group of the images in Fig 7.**

| SD\Image | Eyeball | | Brain | | Forest | | Camouflage car | |
|---|---|---|---|---|---|---|---|---|
| | **Visible1** | **Visible2** | **CT** | **MIR** | **Visible** | **Infrared** | **Visible** | **Infrared** |
| Source map $\sigma$ | 55.83 | 54.64 | 30.92 | **22.269** | **20.45** | 30.32 | 42.83 | 59.81 |
| Grassroots $\sigma$ | 53.83 | 52.83 | **11.49** | **6.961** | **0.32** | **14.59** | **28.78** | 31.92 |

presented as follows:

$$k = 1 + \frac{\sqrt{(\sigma_1 + \sigma_2)} - \sqrt{\sigma_3 + \sigma_4}}{\sqrt{\sigma_3 + \sigma_4}}, \tag{16}$$

A negative correlation is defined by

$$k = 1 - \frac{\sqrt{(\sigma_1 + \sigma_2)} - \sqrt{\sigma_3 + \sigma_4}}{\sqrt{\sigma_3 + \sigma_4}}, \tag{17}$$

where $\sigma_1$ and $\sigma_2$ represent the standard deviations of the two input images, $\sigma_3$ and $\sigma_4$ represent the standard deviations of the $I_{B\max}$ and $I_{B\min}$, respectively.

The ICIF is not limited to specific sensor types, such as visible images and infrared images. Thus, it can be applied to fuse CT images and MRI images, two visible images, and other types of multi-source image fusions, where Eqs (17) and (18) are applied as the fusion weights of the detail layer. The square root of the standard deviation of each source image in Eqs (17) and (18) balances the two source types of images and the two base layer images to the gradient level of the pixel values when compressed as one image. The gradient value here can be indirectly replaced by the standard deviation or other image quality index (such as spatial frequency domain, and average gradient) to select weights adaptively. Fig 8 depicts the results of the fusion.

The objective evaluation corresponding to the fusion results in Fig 8 is shown in Table 2.

Table 2 summarizes the objective evaluation of different fusion strategies. The $Q^{AB/F}$ indicates the number of fused images [56]. A larger value indicates that the fused image contains more information about the source image. $L^{AB/F}$ indicates the loss function [56]. The smaller the value is, the less amount of missing information in the fusion image would be. $N^{AB/F}$ represents the artifacts generated during the fusion process [56]. The smaller the value is, the fewer the artifacts generated in the fused image would be. According to the analysis of Fig 8 and Table 2, the weight fusion strategy of Eq (17) performs well in the fusion quantity $Q^{AB/F}$, and most of the values far exceed the rest of the fusion strategy

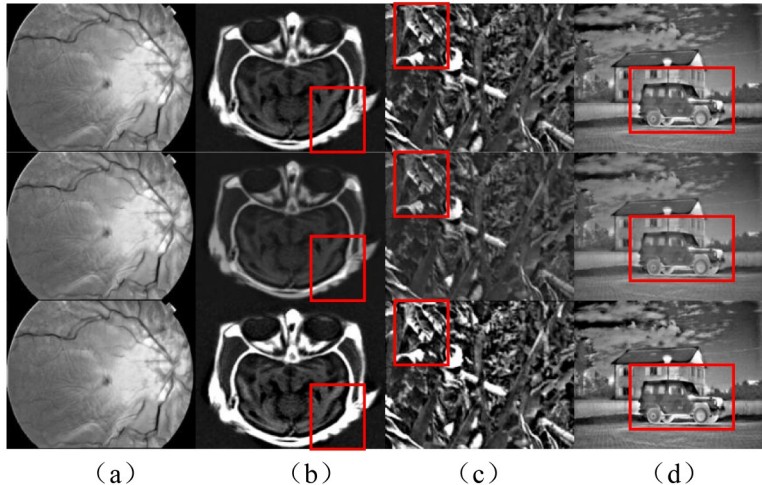

(a)  (b)  (c)  (d)

**Fig 8. Different weights of the detail layer.** The first row represents the fusion result when the fusion weight is 1. The second row represents the fusion result with the fusion weight Eq (16). and The third row represents the fusion result with the fusion weight Eq (17).

**Table 2. Objective evaluation parameters of the fusion results in Fig 8.**

| Image name\Evaluation function | | Eyeball | Forest | Brain | Camouflage car |
|---|---|---|---|---|---|
| $Q^{AB/F}$ | Weight 1 | **0.3149** | 0.3427 | 0.3914 | 0.2482 |
| | Eq (17) | **0.3149** | **0.6019** | **0.7262** | **0.3330** |
| | Eq (16) | 0.3069 | 0.1663 | 0.1835 | 0.1835 |
| $L^{AB/F}$ | Weight 1 | 0.0970 | **0.1145** | 0.1355 | 0.1390 |
| | Eq (17) | 0.0970 | 0.1508 | **0.1139** | 0.1759 |
| | Eq (16) | **0.0958** | 0.1558 | 0.2366 | **0.1305** |
| $N^{AB/F}$ | Weight 1 | **0.1983** | 0.3329 | 1.0815 | 0.2279 |
| | Eq (17) | 0.2007 | **0.1219** | **0.6448** | **0.1639** |
| | Eq (16) | 0.2037 | 0.5445 | 1.1973 | 0.2969 |

weights. In the loss function $L^{AB/F}$, only the median value of the 'Brain' fusion image is found to be the smallest, but when compared with other weight fusion strategies, the difference in values on different test images is found to be smaller. In the evaluation of the artifacts produced by the fusion image, the weight fusion strategy of Eq (17) only performs slightly worse in the 'Eyeball' image fusion, and the rest of the test images produce the least artifacts. Therefore, it is indicated that the fusion weight strategy of Eq (17) is the most robust in objective evaluation. As shown in Fig 8, the weight fusion strategy of Eq (17) has more robustness in various image fusion processes. However, the weight fusion strategy of Eq (16) stands out in image texture when some fused images based on a subjective visual evaluation are under consideration.

When Fig 8 is closely examined, Tables 1 and 2 show that the images with better subjective visual effects using Eq (16) are 'Eyeball' and 'Camouflaged cars,' and the images with poorer subjective visual effects are 'Woods' and 'Brains.' Since the weight is too large, the environment and the target are both enhanced which affects the recognition of the target in the fusion image. The fusion effect of Eq (17) is the opposite of that of Eq (16). In the 'Forest' and the 'Brain,' the target is identified, while in the images of 'Camouflaged cars' and 'Eyeball,' the target is weaker. For each source image feature in Table 2, the standard deviation of at least one image in the two source types of images with better subjective effect using Eq (16) is less than 30. This result implies that this paper proposes an adaptive weight fusion scheme that adjusts the weights adaptively according to the sum of the square root of the information between the two source types of images and the maximum and minimum base information. In this way, the information from the two source types of images complements and cancels each other, resulting in the amount of information obtained being less than the sum of the two images. Therefore, to ensure algorithm efficiency, we use a squared extraction to fit the amount of information in the offset part. When the detail layer is merged with the base layer, the purpose of excessive enhancement will not appear.

We define the implementation conditions for different types of image fusion strategies as follows

Condition 1: $\{\sigma_1 > 30 \; and \; \sigma_2 > 30\}$,

Condition 2: $\{\sigma_1 < 30 \; or \; \sigma_2 < 30\}$,

where $\sigma_1$ and $\sigma_2$ represent the standard deviations of the two input images.

The conditions (1) and (2) are designed to allow the algorithm to select the fusion strategy of the detail layer image adaptively, and the detail layer fusion strategy with better visual effects is required, i.e., as shown in Eq (18).

The coefficient $k$ is unified into the same expression as follows:

$$k = 1 + \frac{\min(\sigma_1, \sigma_2) - 30}{\min(\sigma_1, \sigma_2) - 30} \times \frac{\sqrt{(\sigma_1 + \sigma_2)} - \sqrt{\sigma_3 + \sigma_4}}{\sqrt{\sigma_3 + \sigma_4}}, \tag{18}$$

where $\sigma_3$ and $\sigma_4$ represent the standard deviations of the $I_{B\max}$ and $I_{B\min}$, respectively.

The summary of the proposed algorithm is presented in Table 3.

The computational complexity of the proposed algorithm mainly includes:

1. Multi-scale extraction of the source image is conducted, and the computational complexity is $O(n)$.

2. The basic level of each source map is fused, and the calculation complexity is $O(n^2)$

3. The image eigenvalues of each layer are calculated, and each layer is fused to obtain the fused image. The calculation complexity is $O(n^2)$. The temporal complexity of the ICIF is denoted by

$$T(n) \rightarrow O(n) + O(n^2) + O(n^2), \tag{19}$$

The technical route of the proposed algorithm is shown in Fig 9.

## Results and discussion

The experiment was conducted on Windows 10 OS, Intel(R) Core(TM) i7-6700HQ CPU @ Dual-Core 2.60GHz, and 8GB RAM, and all the algorithms were implemented with MATLAB2016a.

In this section, the performance of the proposed method ICIF is verified with qualitative and quantitative evaluations. The compared methods include MDLatLRR [17], ResNetFusion [35], GANMcC [35], NestFuse [39], SEDRFuse [37], STDFusionNet [36], FusionGAN [45], RTVD-VIF [57], and the MMIF [58]. To ensure an objective evaluation, the following evaluation indexes are used, namely, average gradient (AG) [59], information entropy (H) [60], standard deviation (SD) [61], spatial frequency (SF) [62], edge strength (EI) [63], fusion loss function ($L^{AB/F}$) [56], fusion volume function ($Q^{AB/F}$) [56], and the artifact function ($N^{AB/F}$) [56]. Eq (18) is adopted as the detail-level fusion strategy. To assure a fair comparison, all the algorithms were preprocessed with the proposed enhancement strategy. The fusion performance of the proposed ICIF is verified on infrared and visible light images in the TON database. Then, it is also verified for the other fusion tasks, including multi-focus images, medical images, and other fields.

**Table 3.**

| ICIF: Image fusion via information clustering and image features |
| --- |
| **Input:** Source image $I_1$, Source image $I_2$. |
| **Output:** Fusion image $I_F$ |
| 1. The multi-scale decomposition tool *IC* and Eqs (10) and (11) are used to decompose the input image to obtain the base layer, bright detail layer, and dark detail layer. |
| 2. Use Eq (14) to fuse the base image $I_{BF}$ of each source image. |
| 3. Use Eq (15) to fuse the bright detail layer and dark detail layer into $I_{BF}$, and calculate the fusion weight according to Eq (18). Obtain the final fusion result $I_F$. |

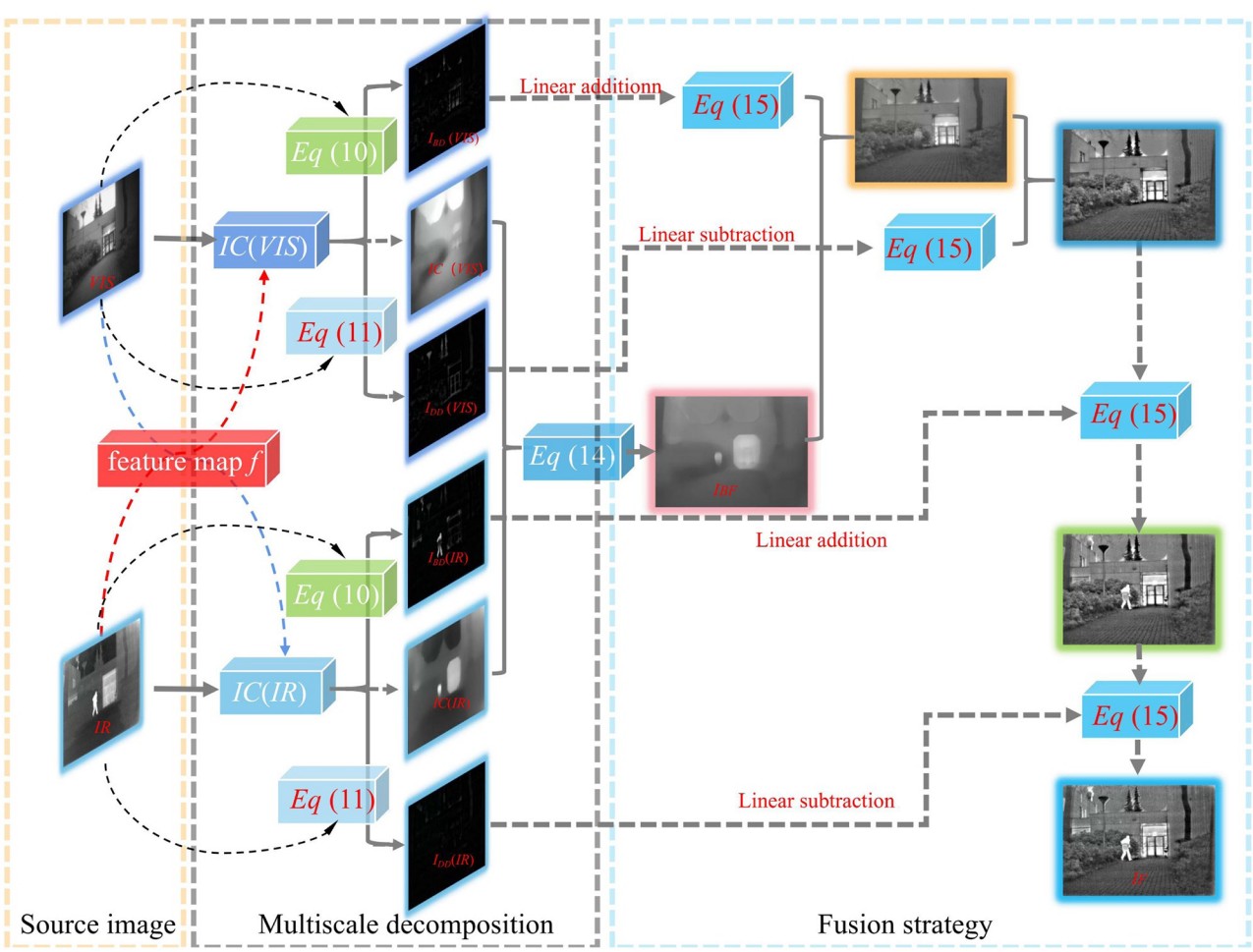

**Fig 9. ICIF technology roadmap.**

## Evaluation metrics

The average gradient (AG) can be used to measure the contrast of small details in an image, which can also reflect the characteristics of texture transformation. The AG measures the clarity of the fused image. Besides, a larger AG value indicates a better image fusion quality. $H$ is defined based on information theory, which measures the amount of information within the fused image. Besides, a larger entropy means more information in the image and better performance of the method. Standard deviation (SD) is a metric reflecting contrast and distribution. The attention of humans is more likely to be directed to the area with high contrast. Accordingly, the larger the SD is, the better the visual effect the fused image would achieve. Spatial frequency (SF) is based on the gradient distribution to effectively reveal the details and texture of the image. The edge intensity (EI) reflects the sharpness of the image. The greater the edge intensity is, the higher the image's sharpness would be. Information fusion quantity would be denoted by ($Q^{AB/F}$) representing a function containing the amount of information about each source graph in the fusion image. The larger the $Q^{AB/F}$ value is, the greater the amount of information the fusion image would contain about each source graph, whose range is contained in $0 < Q^{AB/F} < 1$, where 0 means complete loss of input information, while 1 indicates the ideal

fusion with no loss of input information. Fusion loss ($L^{AB/F}$) indicates the amount of each source graph information lost during fusion, whose smaller value indicates the little information loss. Fusion artifact ($N^{AB/F}$) represents newly generated information during the source graph fusion. A smaller $N^{AB/F}$ represents less newly generated information.

## Contrast experiment

**(i) Qualitative evaluation.** The TNO dataset contains multispectral (such as enhanced vision, near-infrared and long-wave infrared, and thermal) nocturnal images of different military-related scenes. The TNO dataset contains 60 pairs of infrared and visible light images and three sequences involving 19, 32, and 23 image pairs, respectively. We selected Seven of the most challenging and commonly used images (by other researchers) for qualitative evaluation of objects, (http://figshare.com/articles/TNO_Image_Fusion_Dataset/1008029).Noted that all source images were processed with the proposed enhancement strategy to assure a fair comparison. Fig 10 shows the qualitative comparisons of the conducted methods.

Fig 10 depicts that all methods can fuse the information of the visible image and infrared image well to some extent. However, The compared methods, MDLatLRR, ResNetFusion, GANMcC, NestFuse, SEDRFuse, STDFusionNet, FusionGAN, RTVD-VIF, and, MMIF show that the targets (e.g., the building, human, or car) in the fused images are not obvious, indicating less preservation of the thermal radiation information in the infrared images. By contrast, the proposed ICIF can effectively highlight the target area, which is beneficial for target recognition and localization, especially in camouflaged scenes.

Also, experiments on the other sensor fusions were conducted to verify the generalization ability of the proposed ICIF algorithm(http://www.med.harvard.edu/AANLIB/home.html). The fusion results are shown in Fig 11.

Figs 10 and 11 suggest that the proposed method can highlight the target regions better in the fused images than in the visible images. The fusion results of the proposed ICIF contain more abundant detailed information, and our results are more appropriate for human visual perception. For example, Fig 10 depicts that the STDFusionNet and the MMIF fuse only the infrared information of the engine and tires into the visible image, but the clouds in the sky are not integrated into the same image. Also, the fused image by the FusionGAN looks so unnatural. Thus, there is no difference between the fused image and the infrared image for the ResNetFusion and the RTVD-VIF. Other algorithms merge important information, including clouds, engines, and tires from the infrared, into the visible image. When compared to the other methods, the proposed ICIF highlights not only the important target information but also the body of the car and the surrounding environment. Also, the entire image looks more natural. The ResNetFusion fusion image has a tearing phenomenon, and the sidewalk bypass arrow information in the image has completely disappeared. There exist black patches. In the FusionGAN fused image, sidewalk bypass sign arrow information cannot be distinguished. Other algorithms perform well overall. Only ICIF, MDLatLRR, SEDRFuse, and STDFusionNet can reflect the information on the tables and chairs in front of the street shops. The proposed ICIF provides very clear text on the canopy and the texture information of various buildings. More specifically, for the red enlarged area in the experimental image, the analysis shows that the results of the proposed ICIF are the clearest in terms of texture details and heat source targets.

In the extended experiments, the ResNetFusion fusion result has a higher visual similarity with the microwave images. However, the most visible information is lost. The fusion result of STDFusionNet is similar to visible images. However, most visible light and microwave information is lost. Other algorithms can all integrate the information provided by microwave and

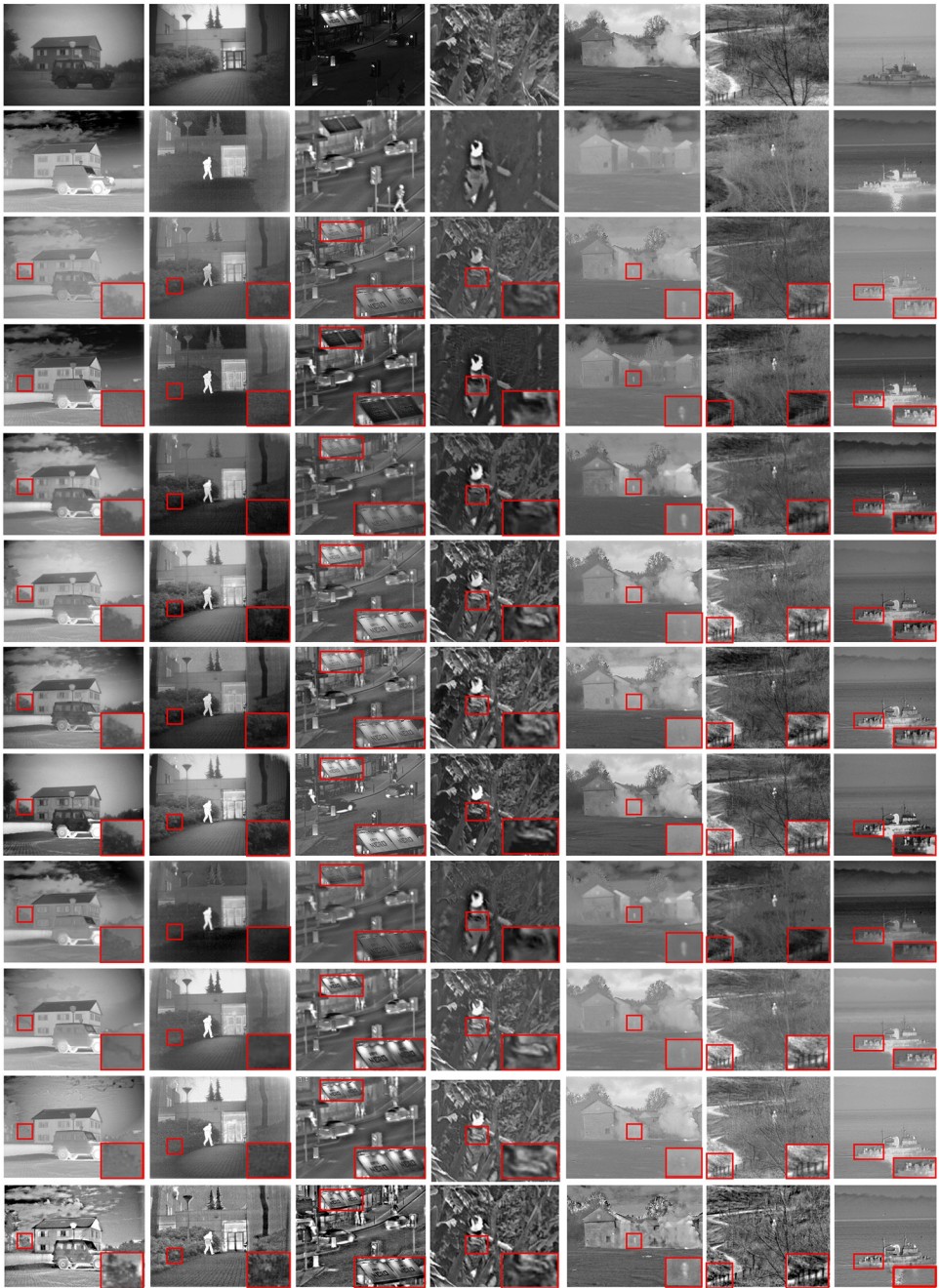

**Fig 10. Qualitative comparisons of seven typical infrared and visible image pairs from the TON database.** From left to right: 'A camouflaged car', 'Doorway', 'Street', 'Forest', 'Smoke', 'Forest trail', 'Fishing boat' (for experiments 1–8) From top to bottom: infrared images, visible images, MDLatLRR, ResNetFusion, GANMcC, NestFuse, SEDRFuse, STDFusionNet, FusionGAN, and the proposed ICIF.

visible light into one image, but the fused images by MDLatLRR, GANMcC, and NestFuse do not show the information of the gun on the chest of the third person clearly. The fused image of the proposed ICIF can not only display the information about the 'Gun' but also rely on the complementary strategy of microwave and visible light image texture to highlight the

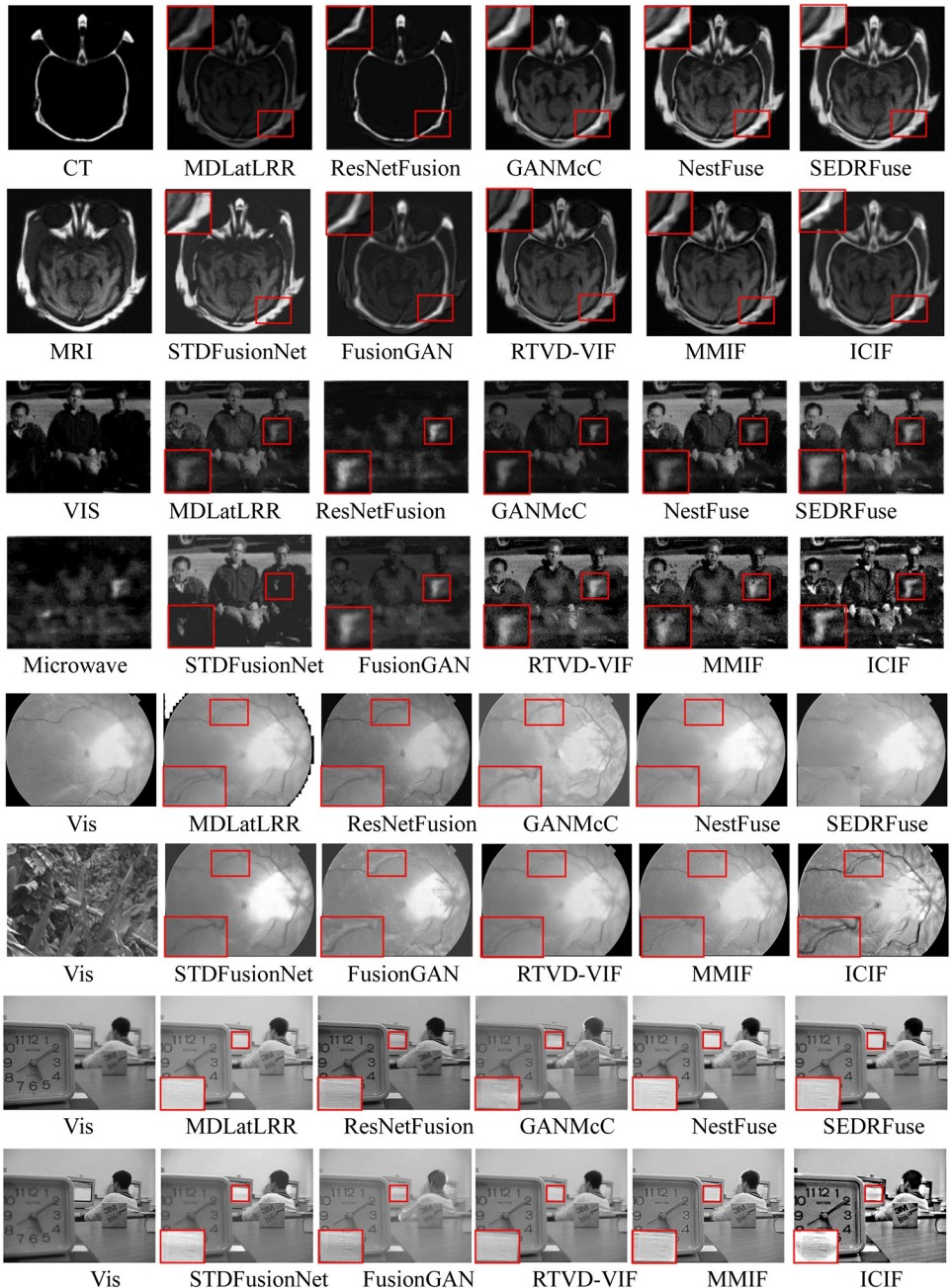

**Fig 11. The generalizability comparison of the other sensor fusion tasks: From top to bottom, 'Brain' 'Gun' 'Eyeball', and 'Lab' (for experiments 9–11).**

information about the character's clothing and the surrounding environment. Furthermore, similar phenomena can also be observed in the other examples, which demonstrates that the proposed ICIF has better performance than the other state-of-the-art in terms of simultaneously preserving thermal radiation information and texture detail information. The extended experiments demonstrate the generalizability of the proposed ICIF. Noted that the

Y-axis of each table in Fig 11 represents the evaluation function value, and the x-axis represents experiments 1–8 in Fig 10 and Experiments 9–11 in Fig 11.

**(ii) Quantitative evaluation.** The following conclusions are drawn with the results of each fusion algorithm based on the employed eight objective evaluation indexes. The fusion results are shown in Fig 12.

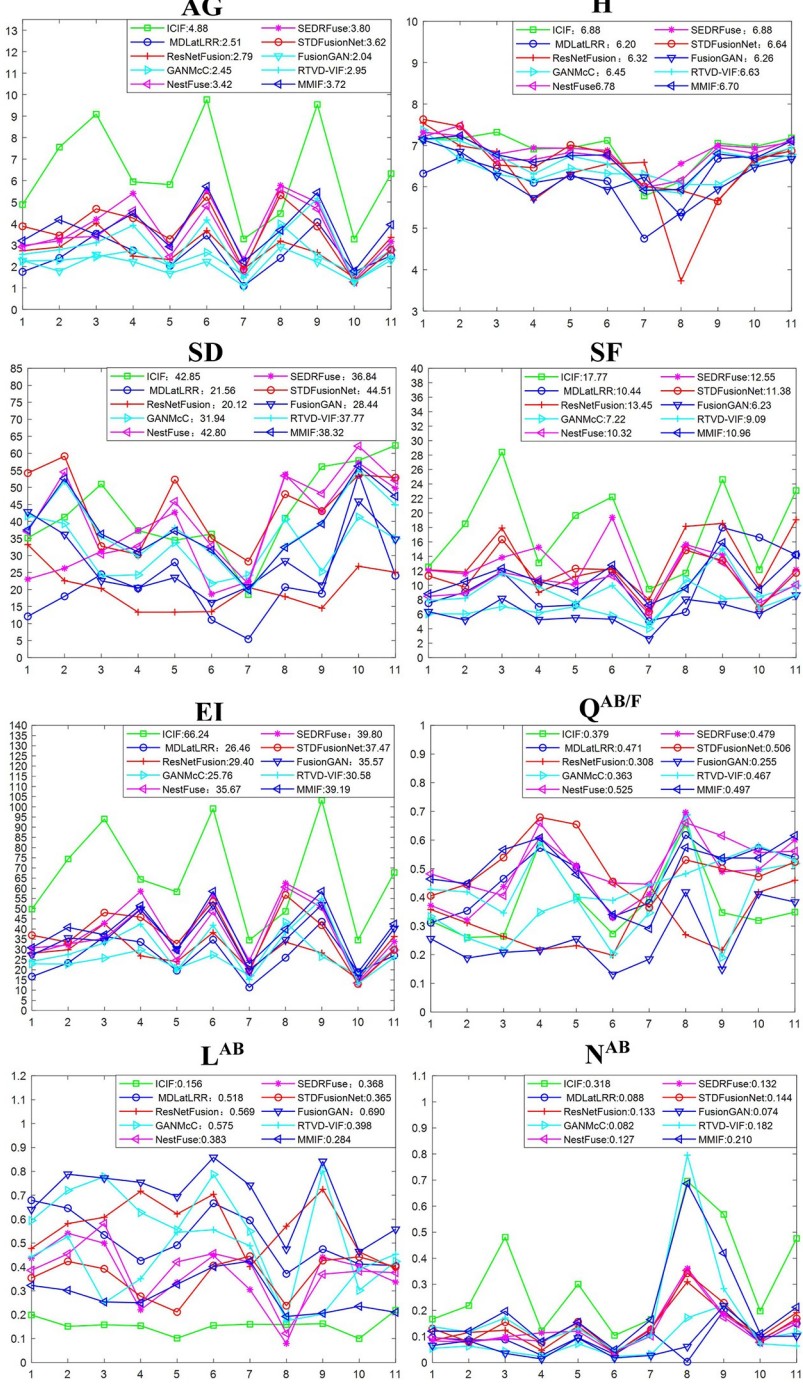

**Fig 12. Objective evaluation results in terms of various image quality indexes.** The abscissa represents the experimental serial number and the ordinate represents the evaluation value.

1. The larger average AG value of the proposed ICIF indicates that the image clarity of our fusion result is higher.

2. The H average value of the proposed ICIF is at the forefront, indicating that the sink redundancy of our fusion result is better.

3. The SD average value of the proposed ICIF is at the forefront, indicating that the visual effect of our result is better.

4. The SF average value of the proposed ICIF is larger, indicating that our fusion result has excellent texture detail.

5. The larger EI average value of the proposed ICIF indicates that our fusion result has excellent sharpness.

6. The larger $Q^{AB/F}$ average value of the proposed ICIF indicates that our fusion result retains the complete information of the source image. The reason why the evaluation value is not optimal is, (*i*) the other comparison algorithms are also the most advanced methods at present. (*ii*) ICIF fuses according to the principle of expanding the texture gradient, which can amplify the signal, resulting in a certain difference from the texture gradient of the source image. Therefore, the evaluation value here is not optimal.

7. The $L^{AB/F}$ average value of the proposed ICIF is smaller, which indicates that our fusion result has the least loss to the source image fusion process.

8. The larger $N^{AB/F}$ average value of the proposed ICIF indicates that there is a difference between our fusion results and the source image since we enhance the weak texture information in the image, making it easier for the image to contain multiple pieces of information to meet the differences caused by subsequent processing.

**(iii) The comparison of computational times.**   Table 4 compares the running time of the comparison methods. MDLatLRR and RTVD-VIF is the most computationally complex algorithm, followed by GANMcC and Nesfuse. The computational time of the proposed ICIF is comparable to other algorithms in all experiments. Noted that the proposed ICIF operated slower than FusionGAN in Experiments 1 and 5, because the two experimental data were used by FusionGAN as training data and, therefore, the data was processed efficiently when used as test data again.

**Table 4. Running time (t/s) of different algorithms in different experiments.**

| Algorithm\ Experiment | 1 | 2 | 3 | 4 | 5 | 6 | 7 | 8 | 9 | 10 | 11 |
|---|---|---|---|---|---|---|---|---|---|---|---|
| ICIF | 1.07 | **0.95** | **1.13** | **0.59** | 1.09 | 1.38 | **0.79** | **0.27** | **0.84** | **1.29** | **1.07** |
| MDLatLRR | 93.8 | 89.2 | 31.63 | 16.3 | 34.2 | 82.4 | 84.2 | 5.38 | 78.3 | 45.4 | 98.8 |
| RseNetFusion | 1.85 | 1.76 | 2.15 | 1.03 | 1.97 | 1.76 | 1.89 | 0.46 | 1.59 | 3.09 | 2.11 |
| GANMcC | 8.94 | 8.84 | 9.41 | 4.72 | 9.94 | 8.87 | 8.69 | 2.12 | 8.36 | 13.5 | 9.35 |
| NestFuse | 19.8 | 19.8 | 22.1 | 8.27 | 20.2 | 20.4 | 12.8 | 23.6 | 20.3 | 30.3 | 22.7 |
| SEDRFuse | 3.31 | 3.29 | 3.64 | 1.91 | 3.63 | 3.17 | 3.06 | 3.49 | 2.92 | 5.22 | 3.52 |
| STDFusionNet | 1.21 | 1.20 | 1.39 | 0.66 | 1.39 | **1.18** | 1.14 | 0.30 | 1.32 | 1.94 | 1.35 |
| RTVD-VIF | 3.82 | 3.95 | 4.39 | 2.18 | 4.37 | 4.01 | 3.79 | 0.76 | 3.63 | 5.43 | 4.18 |
| MMIF | 72.4 | 86.3 | 73.5 | 40.0 | 69.6 | 129.9 | 55.3 | 14.9 | 58.4 | 78.9 | 95.9 |
| FusionGAN | **0.35** | 2.40 | 2.66 | 1.38 | **0.39** | 2.36 | 2.27 | 0.88 | 2.14 | 4.53 | 2.63 |

In conclusion, the proposed algorithm can simultaneously maintain the thermal radiation information in infrared images and the richer texture details in visible light images. When compared to existing state-of-the-art fusion methods, the results appear to have clearer and more prominent goals and richer detailed information. Furthermore, the proposed method has comparably efficient when state-of-the-art techniques are under consideration.

## Conclusion

The manuscript proposes the ICIF algorithm that can be applied to fuse the images collected by employing different sensors, improving the versatility of the fusion algorithm. So, the obtained fused images showed a higher degree of recognition, and objective evaluation indexes were conducive to highlighting the target information and subsequent processing of image engineering. Further, the proposed method provides an adaptive fusion strategy for fusion weights to meet the demand for fusion images in different application scenarios. When compared with other fusion algorithms, the ICIF can fully integrate the information of the source image into an image, which is necessary for important targets.

## Supporting information

**S1 File.**
(DOCX)

**S2 File.**
(DOCX)

## Author Contributions

**Conceptualization:** Linlu Dong.

**Data curation:** Linlu Dong, Jun Wang, Yun Zhang.

**Formal analysis:** Linlu Dong, Jun Wang.

**Funding acquisition:** Jun Wang.

**Investigation:** Linlu Dong, Liangjun Zhao.

**Methodology:** Linlu Dong, Jun Wang, Liangjun Zhao.

**Project administration:** Jun Wang, Liangjun Zhao.

**Resources:** Linlu Dong, Jun Wang, Liangjun Zhao.

**Software:** Linlu Dong, Jun Wang, Jie Yang.

**Supervision:** Linlu Dong, Jun Wang, Liangjun Zhao, Yun Zhang.

**Validation:** Linlu Dong, Jun Wang.

**Visualization:** Linlu Dong, Jun Wang, Yun Zhang.

**Writing – original draft:** Linlu Dong, Jun Wang.

**Writing – review & editing:** Linlu Dong, Jun Wang, Yun Zhang, Jie Yang.

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
