## [Decision Letter · Decision Letter 0]

10 Nov 2022

PONE-D-22-23125ICIF: image fusion via information clustering and image featuresPLOS ONE

Dear Dr. Wang,

Thank you for submitting your manuscript to PLOS ONE. After careful consideration, we feel that it has merit but does not fully meet PLOS ONE’s publication criteria as it currently stands. Therefore, we invite you to submit a revised version of the manuscript that addresses the points raised during the review process.

We look forward to receiving your revised manuscript.

Kind regards,

Ashwani Kumar, Ph.D.

Academic Editor

PLOS ONE

Staff Editor comments:

One or more of the reviewers has recommended that you cite specific previously published works. Members of the editorial team have determined that some of the works referenced are not directly related to the submitted manuscript. As such, please note that it is not necessary or expected to cite the works requested by the reviewer. 

Journal Requirements:

"The authors received no specific funding for this work."

7. Please amend either the abstract on the online submission form (via Edit Submission) or the abstract in the manuscript so that they are identical.

Reviewers' comments:

Reviewer's Responses to Questions

**Comments to the Author**

1. Is the manuscript technically sound, and do the data support the conclusions?

Reviewer #1: Yes

Reviewer #2: Yes

2. Has the statistical analysis been performed appropriately and rigorously? 

Reviewer #1: Yes

Reviewer #2: Yes

3. Have the authors made all data underlying the findings in their manuscript fully available?

Reviewer #1: Yes

Reviewer #2: Yes

4. Is the manuscript presented in an intelligible fashion and written in standard English?

Reviewer #1: Yes

Reviewer #2: No

5. Review Comments to the Author

Reviewer #1: I have now completed the review of the manuscript titled " ICIF: image fusion via information clustering and image features”. This paper proposes a fusion method via information clustering and image features. First, the preprocessed image is decomposed into a basic layer, a bright detail layer, and a dark detail layer using image region clustering. The fuzzy fusion principle is used at the base layer to separate the maximum base image and the minimum base image.

Then, adaptive fusion weights are computed according to the feature value of the image. According to the principle of extending texture details, the fused image is reconstructed. I have some suggestions to further improve the quality of the manuscript.

1. The introduction sections starts with fusion and its demand then it goes to training algorithms, I suggest authors should start with ML/DL area, then what it offers like conventional and training based models.

2. Introduction section should be supported with research in ML/ DL area for various applications to make it more interesting and complete, add the following paragraph and works in the introduction section as second paragraph then add sentiment analysis with this paragraph for a joined second paragraph. i.e.

“Nowadays, scientists and researchers used the machine learning (ML) and Deep learning (DL) models in several applications including image fusion [1-2], agriculture [3, 4], environment [5-11], text sentiment analyses [12], medicine [13], cyber security [14-16]”

3. Section 3.3 “the FFOA can be employed in this paper”, this should write as “the FFOA is employed in this paper”

4. Please add the machine information in the computational complexity of the models, see [3-5].

5. The intra comparison of the proposed models is performed while the inter-comparison or comparison with other studies is missing, please add.

Overall the paper is interesting however further suggested improvement is required to improve the overall quality of the manuscript.

References

1. Fusion-Based Deep Learning Model for Hyperspectral Images Classification

2. Multimodal Medical Supervised Image Fusion Method by CNN

3. Planetscope Nanosatellites Image Classification Using Machine Learning

4. CNN Based Automated Weed Detection System Using UAV Imagery

5. SMOTEDNN: A Novel Model for Air Pollution Forecasting and AQI Classification

6. CDLSTM: A Novel Model for Climate Change Forecasting

7. Deep Learning Based Modeling of Groundwater Storage Change

8. Deep Learning Based Supervised Image Classification Using UAV Images for Forest Areas Classification

9. Bulk Processing of Multi-Temporal Modis Data, Statistical Analyses and Machine Learning Algorithms to Understand Climate Variables in the Indian Himalayan Region

10. Study of permafrost distribution in Sikkim Himalayas using Sentinel-2 satellite images and logistic regression modelling

11. Efficiency of artificial neural networks for glacier ice-thickness estimation: A case study in western Himalaya, India

12. Sentiment analysis using machine learning: Progress in the machine intelligence for data science

13. Fine‑tuned convolutional neural network for different cardiac view classification

14. Insider Threat Detection Based on NLP Word Embedding and Machine Learning

15. DNNBoT: Deep Neural Network-Based Botnet Detection and Classification

16. Development of PCCNN-Based Network Intrusion Detection System for EDGE Computing

Reviewer #2: 1) Please check the English grammar and sentences

2) Why the author is siting only one most recent paper in 2021 ? Why not more ? What about 2022 ?

3) The images are not clearly visible

4) Please check the paper format and spaces

6. PLOS authors have the option to publish the peer review history of their article (what does this mean?). If published, this will include your full peer review and any attached files.

Reviewer #1: No

Reviewer #2: **Yes: **krishna Mohan kudiri

---

## [Author Response · Author response to Decision Letter 0]

14 Dec 2022

Reviewer #1: 

Dear reviewer! Your opinion on the application of MI and DL to image processing is very interesting to us. Although there is a deviation between your opinion and the research content of this paper at this stage, it can better enlighten our next work. We will try our best to reply to your comments. If the reply is not satisfactory, we hope you can give us another chance to modify it until you are satisfied. Because it is very important for us to publish this work on PLOS ONE.

1.The introduction sections starts with fusion and its demand then it goes to training algorithms, I suggest authors should start with ML/DL area, then what it offers like conventional and training based models.

Our actions：We have, in paragraph 4 of the introduction, combined with the question 2 suggestions, supplemented the relevant algorithms for machine learning and deep learning in the field of image fusion, making the introduction more interesting

2. Introduction section should be supported with research in ML/ DL area for various applications to make it more interesting and complete, add the following paragraph and works in the introduction section as second paragraph then add sentiment analysis with this paragraph for a joined second paragraph. i.e.

“Nowadays, scientists and researchers used the machine learning (ML) and Deep learning (DL) models in several applications including image fusion [1-2], agriculture [3, 4], environment [5-11], text sentiment analyses [12], medicine [13], cyber security [14-16]”

Our actions：We have added your suggestions to the manuscript to enrich the introduction and references.

Besides, the title called deep-learning methods include both Machine learning (ML) and Deep learning (DL) models that have been broadly implemented in many areas such as image fusion [27,28], agricultural surveillance [29, 30], environmental monitoring [31-37], sentiment analyses [38], medical image processing [39], and cyber security [40-42]. 

3.Section 3.3 “the FFOA can be employed in this paper”, this should write as “the FFOA is employed in this paper”

Our actions：We did not find corresponding statements in the paper, but we fully revised the language of the manuscript. Hope to satisfy you.For the revision results, see the 'Manuscript'

4.Please add the machine information in the computational complexity of the models, see [3-5].

Our actions：We carefully analyzed the complexity of our algorithm, as shown in Equation (19). At present, our algorithm does not involve machine information, but we will refer to the provided literature in future research [1~16], so as to allow machine learning to play an important role in the field of image fusion.

The computational complexity of the proposed algorithm mainly includes:

(1)Multi-scale extraction of the source image is conducted, and the computational complexity is. 

(2)The basic level of each source map is fused, and the calculation complexity is 

(3)The image eigenvalues of each layer are calculated, and each layer is fused to obtain the fused image. The calculation complexity is. The temporal complexity of the ICIF is denoted by

 , (19)

5.The intra comparison of the proposed models is performed while the inter-comparison or comparison with other studies is missing, please add.

Our actions：We analyzed the differences between our method and other research methods in the 'motivational section', further demonstrating the contribution of this article.

To clearly express the novel points of the proposed approach, we use a viewable approach to show the relevant work (ADF [54]) and how the proposed method differs is presented. Among them, the clustering of the ADF is shown in Fig 2 (a), and the proposed clustering process is shown in Fig 2 (b). 

(a)Clustering results and technical process of the ADF algorithm

(b) Clustering results and technical flow of the proposed algorithm

Fig 2. Comparison of the existing art and the proposed technology

To further dissect the difference between the proposed technique and the available techniques, the surf plots of some regions of the clustering results based on the 'AD base image' and 'IC base image' in Fig 2 are plotted separately as shown in Fig 3.

(a)Infrared image (b) AD base image (c) IC base image

Fig 3. Surf map of some regions of image clustering results. (a) The upper row represents the infrared image, and the lower row represents the surf map of the corresponding area of the green rectangular box. (b) The upper row shows the clustering effect of the anisotropic diffusion (AD) method, and the lower row shows the surf map of the corresponding area of the green rectangular box. (c) The upper row shows the clustering effect of our clustering (IC) method, and the lower row shows the surf map of the corresponding area of the green rectangular box. While the X-axis represents the row coordinates of the pixel points in the image (set value range: 0 ~ 100), the Y-axis represents the column coordinates of the pixel points in the image (set value range: 0 ~ 100), and the Z-axis represents the values of the pixel points corresponding to each coordinate point (value range is determined by the source image)

Fig 3 depicts a surf plot of the local region of the infrared image, with most of the texture details highlighted on the surface. In the surf diagram of the local region of the AD base image, although the surface is smooth, there is also a large amount of bulge information between the peak and the trough, such as at the green circular box. The corresponding AD base image also retains a large amount of significant target information, such as the human leg information at the red arrow, which indicates that this part of the information in the decomposed detail feature layer has not been extracted. When the clustering process is considered, the proposed method adopts the texture feature of the visible light image as the guide map, so that the prominent texture at the green circular box in the surf plot in Fig 3(c) is smoothed because the low texture gradient at the corresponding position of the visible light image in Fig 3 expands the degree of smoothing, which improves the correlation between the infrared and the visible light images regarding feature extraction process. Therefore, the detailed feature layer obtained after the decomposition contains this partial information.

Overall the paper is interesting however further suggested improvement is required to improve the overall quality of the manuscript.

References

1.Fusion-Based Deep Learning Model for Hyperspectral Images Classification

2. Multimodal Medical Supervised Image Fusion Method by CNN

3. Planetscope Nanosatellites Image Classification Using Machine Learning

4. CNN Based Automated Weed Detection System Using UAV Imagery

5.SMOTEDNN: A Novel Model for Air Pollution Forecasting and AQI Classification

6.CDLSTM: A Novel Model for Climate Change Forecasting

7.Deep Learning Based Modeling of Groundwater Storage Change

8. Deep Learning Based Supervised Image Classification Using UAV Images for Forest Areas Classification

9.Bulk Processing of Multi-Temporal Modis Data, Statistical Analyses and Machine Learning Algorithms to Understand Climate Variables in the Indian Himalayan Region

10.Study of permafrost distribution in Sikkim Himalayas using Sentinel-2 satellite images and logistic regression modelling

11. Efficiency of artificial neural networks for glacier ice-thickness estimation: A case study in western Himalaya, India

12.Sentiment analysis using machine learning: Progress in the machine intelligence for data science

13.Finetuned convolutional neural network for different cardiac view classification

14. Insider Threat Detection Based on NLP Word Embedding and Machine Learning

15. DNNBoT: Deep Neural Network-Based Botnet Detection and Classification

16. Development of PCCNN-Based Network Intrusion Detection System for EDGE Computing

Reviewer #2: 

1) Please check the English grammar and sentences

Our actions：We submitted the paper to the language modification agency, and the editor of the native English language has comprehensively revised the grammar and sentences of the paper.

2)Why the author is siting only one most recent paper in 2021 ? Why not more ? What about 2022 ?

We previously understood the error as one to supplement the relevant work of other researchers in 2022. So in the experimental section, we added several recent experiments to verify the advantages of our algorithm and remade the line plots. See the experiment section, where we found that the problem involves other results we have published. The reason is that most of our work in 2022 has been accepted by journals, but it has not been published yet, and some of the work is still in the external review stage. I hope this answer will satisfy you.

(i) Qualitative evaluation.

 Seven typical image pairs are selected to assess fusion processing qualitatively. Noted that all source images were processed with the proposed enhancement strategy to assure a fair comparison. Fig 10 shows the qualitative comparisons of the conducted methods. 

Fig 10 depicts that all methods can fuse the information of the visible image and infrared image well to some extent. However, The compared methods, MDLatLRR, ResNetFusion, GANMcC, NestFuse, SEDRFuse, STDFusionNet, FusionGAN, RTVD-VIF, and, MMIF show that the targets (e.g., the building, human, or car) in the fused images are not obvious, indicating less preservation of the thermal radiation information in the infrared images. By contrast, the proposed ICIF can effectively highlight the target area, which is beneficial for target recognition and localization, especially in camouflaged scenes.

Fig 10. Qualitative comparisons of seven typical infrared and visible image pairs from the TON database. From left to right: 'A camouflaged car', 'Doorway', 'Street', 'Forest', 'Smoke', 'Forest trail', 'Fishing boat' (for experiments 1 – 8) From top to bottom: infrared images, visible images, MDLatLRR, ResNetFusion, GANMcC, NestFuse, SEDRFuse, STDFusionNet, FusionGAN, and the proposed ICIF.

Also, experiments on the other sensor fusions were conducted to verify the generalization ability of the proposed ICIF algorithm. The fusion results are shown in Fig 11.

 CT MDLatLRR ResNetFusion GANMcC NestFuse SEDRFuse 

 MRI STDFusionNet FusionGAN RTVD-VIF MMIF ICIF

VIS MDLatLRR ResNetFusion GANMcC NestFuse SEDRFuse

 Microwave STDFusionNet FusionGAN RTVD-VIF MMIF ICIF

 Vis MDLatLRR ResNetFusion GANMcC NestFuse SEDRFuse 

Vis STDFusionNet FusionGAN RTVD-VIF MMIF ICIF

 Vis MDLatLRR ResNetFusion GANMcC NestFuse SEDRFuse 

Vis STDFusionNet FusionGAN RTVD-VIF MMIF ICIF

Fig 11. The generalizability comparison of the other sensor fusion tasks: From top to bottom, 'Brain' 'Gun' 'Eyeball', and ‘Lab' (for experiments 9 – 11).

Figs 10 and 11 suggest that the proposed method can highlight the target regions better in the fused images than in the visible images. The fusion results of the proposed ICIF contain more abundant detailed information, and our results are more appropriate for human visual perception. For example, Fig 10 depicts that the STDFusionNet and the MMIF fuse only the infrared information of the engine and tires into the visible image, but the clouds in the sky are not integrated into the same image. Also, the fused image by the FusionGAN looks so unnatural. Thus, there is no difference between the fused image and the infrared image for the ResNetFusion and the RTVD-VIF. Other algorithms merge important information, including clouds, engines, and tires from the infrared, into the visible image. When compared to the other methods, the proposed ICIF highlights not only the important target information but also the body of the car and the surrounding environment. Also, the entire image looks more natural. The ResNetFusion fusion image has a tearing phenomenon, and the sidewalk bypass arrow information in the image has completely disappeared. There exist black patches. In the FusionGAN fused image, sidewalk bypass sign arrow information cannot be distinguished. Other algorithms perform well overall. Only ICIF, MDLatLRR, SEDRFuse, and STDFusionNet can reflect the information on the tables and chairs in front of the street shops. The proposed ICIF provides very clear text on the canopy and the texture information of various buildings. More specifically, for the red enlarged area in the experimental image, the analysis shows that the results of the proposed ICIF are the clearest in terms of texture details and heat source targets.

 In the extended experiments, the ResNetFusion fusion result has a higher visual similarity with the microwave images. However, the most visible information is lost. The fusion result of STDFusionNet is similar to visible images. However, most visible light and microwave information is lost. Other algorithms can all integrate the information provided by microwave and visible light into one image, but the fused images by MDLatLRR, GANMcC, and NestFuse do not show the information of the gun on the chest of the third person clearly. The fused image of the proposed ICIF can not only display the information about the 'Gun' but also rely on the complementary strategy of microwave and visible light image texture to highlight the information about the character's clothing and the surrounding environment. Furthermore, similar phenomena can also be observed in the other examples, which demonstrates that the proposed ICIF has better performance than the other state-of-the-art in terms of simultaneously preserving thermal radiation information and texture detail information. The extended experiments demonstrate the generalizability of the proposed ICIF. Noted that the Y-axis of each table in Fig 11 represents the evaluation function value, and the x-axis represents experiments 1-8 in Fig 10 and Experiments 9-11 in Fig 11.

(ii) Quantitative evaluation

The following conclusions are drawn with the results of each fusion algorithm based on the employed eight objective evaluation indexes.The fusion results are shown in Fig 12.

(1) The larger average AG value of the proposed ICIF indicates that the image clarity of our fusion result is higher.

(2) The H average value of the proposed ICIF is at the forefront, indicating that the sink redundancy of our fusion result is better.

(3) The SD average value of the proposed ICIF is at the forefront, indicating that the visual effect of our result is better.

(4) The SF average value of the proposed ICIF is larger, indicating that our fusion result has excellent texture detail.

(5) The larger EI average value of the proposed ICIF indicates that our fusion result has excellent sharpness. 

(6) The larger QAB/F average value of the proposed ICIF indicates that our fusion result retains the complete information of the source image. The reason why the evaluation value is not optimal is, (i) the other comparison algorithms are also the most advanced methods at present. (ii) ICIF fuses according to the principle of expanding the texture gradient, which can amplify the signal, resulting in a certain difference from the texture gradient of the source image. Therefore, the evaluation value here is not optimal.

(7) The LAB/F average value of the proposed ICIF is smaller, which indicates that our fusion result has the least loss to the source image fusion process.

(8) The larger NAB/F average value of the proposed ICIF indicates that there is a difference between our fusion results and the source image since we enhance the weak texture information in the image, making it easier for the image to contain multiple pieces of information to meet the differences caused by subsequent processing.

Fig 12. Objective evaluation results in terms of various image quality indexes. The abscissa represents the experimental serial number and the ordinate represents the evaluation value

(iii) The comparison of computational times

Table 4. Running time (t/s) of Different Algorithms in Different Experiments

Algorithm\\

Experiment 1 2 3 4 5 6 7 8 9 10 11

ICIF 1.07 0.95 1.13 0.59 1.09 1.38 0.79 0.27 0.84 1.29 1.07

MDLatLRR 93.8 89.2 31.63 16.3 34.2 82.4 84.2 5.38 78.3 45.4 98.8

RseNetFusion 1.85 1.76 2.15 1.03 1.97 1.76 1.89 0.46 1.59 3.09 2.11

GANMcC 8.94 8.84 9.41 4.72 9.94 8.87 8.69 2.12 8.36 13.5 9.35

NestFuse 19.8 19.8 22.1 8.27 20.2 20.4 12.8 23.6 20.3 30.3 22.7

SEDRFuse 3.31 3.29 3.64 1.91 3.63 3.17 3.06 3.49 2.92 5.22 3.52

STDFusionNet 1.21 1.20 1.39 0.66 1.39 1.18 1.14 0.30 1.32 1.94 1.35

RTVD-VIF 3.82 3.95 4.39 2.18 4.37 4.01 3.79 0.76 3.63 5.43 4.18

MMIF 72.4 86.3 73.5 40.0 69.6 129.9 55.3 14.9 58.4 78.9 95.9

FusionGAN 0.35 2.40 2.66 1.38 0.39 2.36 2.27 0.88 2.14 4.53 2.63

Table 4 compares the running time of the comparison methods. MDLatLRR and RTVD-VIF is the most computationally complex algorithm, followed by GANMcC and Nesfuse. The computational time of the proposed ICIF is comparable to other algorithms in all experiments. Noted that the proposed ICIF operated slower than FusionGAN in Experiments 1 and 5, because the two experimental data were used by FusionGAN as training data and, therefore, the data was processed efficiently when used as test data again.

3) The images are not clearly visible

Our actions：The top side represents the image of the original manuscript, and the bottom side represents our modified image, image quality (300 dpi). The other pictures in the paper are the original pictures.

Change to

4) Please check the paper format and spaces

Our actions：We re-typesetting the manuscript, and in the later stage, we will further cooperate with the journal to improve the format of the paper to make our manuscript more perfect.hope it can dispel your concerns about the format.

Thank you very much for your valuable comments, which have played a key role in improving the quality of our manuscript. I hope that our answers will satisfy you. If there are other questions, We hope you can give us another chance to modify them.

---

## [Decision Letter · Decision Letter 1]

21 Feb 2023

PONE-D-22-23125R1ICIF: image fusion via information clustering and image featuresPLOS ONE

Dear Dr. Wang,

Thank you for submitting your manuscript to PLOS ONE. After careful consideration, we feel that it has merit but does not fully meet PLOS ONE’s publication criteria as it currently stands. Therefore, we invite you to submit a revised version of the manuscript that addresses the points raised during the review process. Based on reviewers comments, authors are required to do a minor revision to incorporate suggestions.

We look forward to receiving your revised manuscript.

Kind regards,

Rajiv Singh

Academic Editor

PLOS ONE

Journal Requirements:

Reviewers' comments:

Reviewer's Responses to Questions

**Comments to the Author**

1. If the authors have adequately addressed your comments raised in a previous round of review and you feel that this manuscript is now acceptable for publication, you may indicate that here to bypass the “Comments to the Author” section, enter your conflict of interest statement in the “Confidential to Editor” section, and submit your "Accept" recommendation.

Reviewer #1: All comments have been addressed

Reviewer #2: All comments have been addressed

2. Is the manuscript technically sound, and do the data support the conclusions?

Reviewer #1: Yes

Reviewer #2: Yes

3. Has the statistical analysis been performed appropriately and rigorously? 

Reviewer #1: Yes

Reviewer #2: Yes

4. Have the authors made all data underlying the findings in their manuscript fully available?

Reviewer #1: Yes

Reviewer #2: Yes

5. Is the manuscript presented in an intelligible fashion and written in standard English?

Reviewer #1: Yes

Reviewer #2: Yes

6. Review Comments to the Author

Reviewer #1: Dear Authors

I have now completed the review of the revised manuscript, titled " ICIF: image fusion via information clustering and image features”. I have observed that the authors put in good efforts to address most of the comments satisfactorily.

Best wishes

Reviewer #2: This paper is written very well, but I have few comments for the author

1. Please check your equation format. For example, Equation 14. It has ". I Bmax" and "I Bmin". Please check the difference.

2. Please check your results and discussion. Windows 10 OS, 2.60GHz CPU and 8 GB RAM. What about processor and core ? It is also important.

3. Please check your Table format. For instance, Table 1 size.

4. Please discuss more about which database you have used.

7. PLOS authors have the option to publish the peer review history of their article (what does this mean?). If published, this will include your full peer review and any attached files.

Reviewer #1: No

Reviewer #2: **Yes: **Krishna Mohan Kudiri*

---

## [Author Response · Author response to Decision Letter 1]

22 Feb 2023

Journal Requirements:

Please review your reference list to ensure that it is complete and correct. If you have cited papers that have been retracted, please include the rationale for doing so in the manuscript text, or remove these references and replace them with relevant current references. Any changes to the reference list should be mentioned in the rebuttal letter that accompanies your revised manuscript. If you need to cite a retracted article, indicate the article’s retracted status in the References list and also include a citation and full reference for the retraction notice.1

Our actions：Through checking the references, the above problems are not found.

Reviewer #1: 

Reviewer #2: This paper is written very well, but I have few comments for the author

1. Please check your equation format. For example, Equation 14. It has ". I Bmax" and "I Bmin". Please check the difference.

Our actions：We drew inferences from one example and checked all the equations. And correct the problematic equation.

 , (14)

instead 

 , (14)

2. Please check your results and discussion. Windows 10 OS, 2.60GHz CPU and 8 GB RAM. What about processor and core ? It is also important.

Our actions：We have supplemented this part of information as follows:

The experiment was conducted on Windows 10 OS, Intel(R) Core(TM) i7-6700HQ CPU @ Dual-Core 2.60GHz, and 8GB RAM, and all the algorithms were implemented with MATLAB2016a.

3. Please check your Table format. For instance, Table 1 size.

Our actions：We checked the table, and found an abnormal length of Table 1, and we reworked Table 1

SD\\Image Eyeball Brain Forest Camouflage car

 Visible1 Visible2 CT MIR Visible Infrared Visible Infrared

Source map 55.83 54.64 30.92 22.269 20.45 30.32 42.83 59.81

Grassroots 53.83 52.83 11.49 6.961 0.32 14.59 28.78 31.92

instead 

SD\\Image Eyeball Brain Forest Camouflage car

 Visible1 Visible2 CT MIR Visible Infrared Visible Infrared

Source map 55.83 54.64 30.92 22.269 20.45 30.32 42.83 59.81

Grassroots 53.83 52.83 11.49 6.961 0.32 14.59 28.78 31.92

4. Please discuss more about which database you have used.

Our actions：We introduce the TON database and present links to the database with the aim of improving the availability of data, which are also commonly used by other scholars.

 The TNO dataset contains multispectral (such as enhanced vision, near-infrared and long-wave infrared, and thermal) nocturnal images of different military-related scenes. The TNO dataset contains 60 pairs of infrared and visible light images and three sequences involving 19, 32, and 23 image pairs, respectively. We selected Seven of the most challenging and commonly used images (by other researchers) for qualitative evaluation of objects, (http://figshare.com/articles/TNO_Image_Fusion_Dataset/1008029).

Also, experiments on the other sensor fusions were conducted to verify the generalization ability of the proposed ICIF algorithm(http://www.med.harvard.edu/AANLIB/home.html). The fusion results are shown in Fig 11

---

## [Decision Letter · Decision Letter 2]

7 May 2023

ICIF: image fusion via information clustering and image features

PONE-D-22-23125R2

Dear Dr. Wang,

We’re pleased to inform you that your manuscript has been judged scientifically suitable for publication and will be formally accepted for publication once it meets all outstanding technical requirements.

Kind regards,

Muhammad Shahid Farid, Ph.D.

Academic Editor

PLOS ONE

Additional Editor Comments (optional):

Reviewers' comments:

Reviewer's Responses to Questions

**Comments to the Author**

1. If the authors have adequately addressed your comments raised in a previous round of review and you feel that this manuscript is now acceptable for publication, you may indicate that here to bypass the “Comments to the Author” section, enter your conflict of interest statement in the “Confidential to Editor” section, and submit your "Accept" recommendation.

Reviewer #1: All comments have been addressed

2. Is the manuscript technically sound, and do the data support the conclusions?

Reviewer #1: Yes

3. Has the statistical analysis been performed appropriately and rigorously? 

Reviewer #1: Yes

4. Have the authors made all data underlying the findings in their manuscript fully available?

Reviewer #1: Yes

5. Is the manuscript presented in an intelligible fashion and written in standard English?

Reviewer #1: Yes

6. Review Comments to the Author

Reviewer #1: I have now completed the review of the revised manuscript, titled " ICIF: image fusion via information clustering and image features”. I have observed that the authors put in good efforts to address all the comments satisfactorily.

7. PLOS authors have the option to publish the peer review history of their article (what does this mean?). If published, this will include your full peer review and any attached files.

Reviewer #1: No

---

## [Editor Report · Acceptance letter]

20 Jul 2023

PONE-D-22-23125R2 

ICIF: Image fusion via information clustering and image features 

Dear Dr. Wang:

I'm pleased to inform you that your manuscript has been deemed suitable for publication in PLOS ONE. Congratulations! Your manuscript is now with our production department. 

Kind regards, 

on behalf of

Dr. Muhammad Shahid Farid 

Academic Editor

PLOS ONE